# Effects of Long-Term Low-Protein Diets Supplemented with Sodium Dichloroacetate and Glucose on Metabolic Biomarkers and Intestinal Microbiota of Finishing Pigs

**DOI:** 10.3390/ani12192522

**Published:** 2022-09-21

**Authors:** Yetong Xu, Huiyuan Chen, Ke Wan, Zhiru Tang, Weizhong Sun, Liuting Wu, Zhongxiang Ren, Qi Ding, Kaiyang Liang, Zhihong Sun

**Affiliations:** Laboratory for Bio-Feed and Molecular Nutrition, Department of Animal Science and Technology, Southwest University, Chongqing 400715, China

**Keywords:** low-protein diet, sodium dichloroacetate, metabolic markers, intestinal microbiota, pigs

## Abstract

**Simple Summary:**

Our previous study showed that low protein (LP) diets can save protein sources and reduce nitrogen excretion without significantly impacting the growth performance of growing-finishing pigs. In the current study, sodium dichloroacetate (DCA) was used to explore whether it could work like a metabolic modulator to improve glucose and lipid oxidation in pigs fed with an LP diet, whilst glucose (GLUC) was used to evaluate if it could act as pyruvate, which is a metabolic fuel. In particular, we investigated the effects of LP diets supplemented with DCA and GLUC on metabolic responses and intestinal microbiota of finishing pigs. Our findings showed that DCA and GLUC supplementation improved the hepatic lipid metabolism but had limited effects on improving colonic microbiota profile of finishing pigs.

**Abstract:**

The objective of this study was to evaluate the effects of low-protein (LP) diets supplemented with sodium dichloroacetate (DCA) and glucose (GLUC) on metabolic markers and intestinal microbiota of finishing pigs. A total of 80 crossbred growing barrows were allocated randomly to one of the five treatments, including the normal protein level diet (CON), the LP diets, LP with 120 mg/kg DCA (LP + DCA) or 1.8% glucose (LP + GLUC), and LP with 120 mg/kg DCA and 1.8% glucose (LP + DCA + GLUC). The LP diet increased the plasma HDL, triglyceride, and cholesterol concentrations and reduced the bile acid, urea nitrogen, albumin, and total protein concentrations compared to the CON diet (*p* < 0.05). The LP + DCA + GLUC diet reduced the plasma VLDL, triglyceride, and cholesterol concentrations and increased the bile acid concentration compared with the LP diet (*p* < 0.05). Pigs fed the LP + DCA and LP + GLUC diets showed reduced 3-Hydroxy-3-Methylglutaryl-CoA Reductase content and increased Cytochrome P450 Family 7 Subfamily A Member 1 activity of liver compared that of the CON diet (*p* < 0.05). Moreover, the LP diets with or without DCA and GLUC supplementation increased the relative abundance of colonic microbiota related to carbohydrate fermentation in finishing pigs. In conclusion, 120 mg/kg DCA or 1.8% GLUC supplementation in an LP diet modulated the hepatic lipid metabolism of pigs, while the DCA along with GLUC supplementation likely improved the lipid metabolism by stimulating bile acid secretion.

## 1. Introduction

Dietary protein is the basic source of amino acids for livestock; however, increased dietary protein levels and imbalance of amino acid composition result in low utilization rate of proteins and nitrogen pollution [1,2]. The excess dietary protein and amino acids contribute to the proliferation of pathogenic bacteria in the large intestine and leads to the production of harmful metabolites, which can influence the immunity and inflammatory responses in animals [3,4]. Importantly, a previous study found that a moderate reduction of dietary protein levels (13%) improved the microbial community structure in the ileal and colonic segments of adult pigs [5]. Thus, low protein (LP) diets have become more and more popular, especially in the growing-finishing pigs due to their advantages in saving protein sources, reducing nitrogen excretion and improving gut health without significantly affecting growth performance of animals [6,7].

Some researchers have found that LP diets changed protein and energy metabolic response of muscle in growing pigs [1,8]. The LP diet decreased the amount of deamination and transamination of amino acids due to the imbalance of non-essential amino acids and induced a low utilization rate of amino acid for nitrogen retention in pigs [1,9]. Furthermore, it is possible that supplementation with large amounts of crystalline amino acids in the LP diets might negatively affect energy metabolism especially suppress glucose (GLUC) oxidation [1,10]. Therefore, there is a need to explore if there is any energy modulator added into an LP diet to improve the utilization rate of amino acid and the oxidation of GLUC.

Sodium dichloroacetate (DCA) is non-toxic and generally used to treat cancer and lactic acidosis by reactivating aerobic metabolism and inhibiting pyruvate kinase, which is a key enzyme in glycolysis [11,12,13]. Our previous study demonstrated that DCA decreased amino acid metabolism for urea and increased GLUC consumption by modulating the expression of metabolic enzymes in the intestinal epithelial cells of pigs [14]. Moreover, we found 120 mg/kg DCA supplementation in an LP diet improved the growth performance of growing-finishing pigs [7]. Nonetheless, limited data are available concerning the effects of LP diets supplemented with DCA or GLUC on the metabolic biomarkers and intestinal microbiota profile in finishing pigs. Therefore, the objective of the present study was to explore the effects of supplementing DCA or GLUC to an LP diet on the biomarkers of nutrient metabolism and intestinal microbiota profile in finishing pigs.

## 2. Materials and Methods

The experimental diets, animal housing, and experimental design have been described in detail in our previous paper presenting the effects of DCA and GLUC on growth performance and meat quality of growing-finishing pigs [7].

### 2.1. Experimental Diets

Experimental diets in the three growing phases were formulated based on the body weight of pigs, namely, 25–45, 45–65, and 65–95 kg. The five experimental diets were a control diet (CON) wherein the dietary protein levels were 18%, 16.5%, and 15.5% in the three phases, respectively; a low protein diet (LP) with decreased dietary protein levels by 4.5% compared to the CON; an LP diet supplemented with an extra 120 mg/kg DCA (LP + DCA); an LP diet supplemented with 1.8% glucose (LP + GLUC); an LP diet supplemented with both the 120 mg/kg DCA and 1.8% glucose (LP + DCA + GLUC). The doses of DCA and GLUC were chosen based on our previous studies [15]. The diets were based on corn-soybean meal and Lys, Met, Thr, and Trp were supplemented to the LP diets to balance the limiting amino acids and meet the nutrient requirements for pigs (NRC, 2012). The ingredients and nutrient levels of five experimental diets are shown in Appendix A.

### 2.2. Animals and Experimental Design

All procedures used in the animal study were approved by the Institutional Animal Care and Use Committee of Southwest University (Chongqing, China). Pigs fed DCA were handled following the Experimental Animal Handling Procedure of Southwest University (Chongqing, China).

A total of 80 crossbred barrows (Duroc × Landrace × Large White, 70 ± 2 days of age) with initial body weight (BW) 27 ± 0.5 kg were used in the experiment. Sixteen repetitions of one pig each were included in each dietary treatment. The animal handling was described in detail previously. The experimental period lasted for 12 weeks and pigs were housed in individual stainless-steel metabolic cages (1.8 m length × 0.75 m width × 1.2 m height) with a plastic floor. Pigs were provided feed ad libitum and had free access to drink throughout the experiment. The room temperature was controlled by heaters and exhaust fans at 23–28 °C.

### 2.3. Sample Collection

At the end of the experiment, all pigs were fasted overnight and six pigs per treatment were randomly selected to collect blood samples (5 mL) via anterior vena cava into 10 mL heparinized vacutainer tubes (Becton Dickinson Vacutainer Systems, Franklin Lakes, NJ, USA). Samples were centrifuged 3000 × *g* for 10 min at 4 °C (Biofuge 22R; Heraeus, Hanau, Germany), and the plasma was kept at −80 °C until analysis. The euthanasia procedures were performed to collect samples as described previously [7]. The pigs were humanely killed by exsanguination after electrical stunning, then the thoracic cavity and abdominal cavity were opened by a midline incision and segments (1 cm in length) of the mid-ileum (without digesta) were fixed in 10% neutral buffered formalin for subsequent histological measurement. The liver tissues and colonic digesta were collected in sterilized vials, immediately immersed in liquid nitrogen and then stored at −80 °C for subsequent analysis.

### 2.4. Analytical Methods

#### 2.4.1. Metabolic Markers and Hormonal Index in Plasma

Concentrations of high-density lipoprotein (HDL, ml092659), low-density lipoprotein (LDL, ml092641), very low-density lipoprotein (VLDL, ml026119), triacylglycerol (TG, ml092655), total cholesterol (TC, ml092733), glucose (GLUC, ml092722), total bile acids (TBA, ml092739), blood urea nitrogen (BUN, ml092694), albumin (ALB, ml092880), total protein (TP, ml093077), direct bilirubin (DBil, ml092731), total bilirubin (TBil, ml016910), lactate dehydrogenase (LDH, ml092996), pyruvate dehydrogenase (PDH, ml076572), glutamate dehydrogenase (GDH, ml076495), adenosine deaminase (ADA, ml093026), cholinesterase (ChE, ml092914), γ-glutamyl transferase (GGT, ml092630), alkaline phosphatase (ALP, ml092963), alanine transaminase (ALT, ml092635), aspartate transaminase (AST, ml077324), IgA (ml092680), IgG (ml092681), and IgM (ml092683) in plasma of pigs were measured by biochemical reagent test kits from Shanghai ELISA Biotechnology Co., Ltd. (Shanghai, China). Plasma insulin was determined using a commercially available porcine-specific ELISA kit (m002341) and the plasma free triiodothyronine (T3) and thyroxine (T4) were measured with ELISA kit (ml002375) according to the manufacturer’s instructions (Shanghai ELISA Biotechnology Co., Ltd., Shanghai, China). Measurement of growth hormone (ml002349), glucagon (ml022730), leptin (ml002355) and melanin (ml002425) was conducted by using ELISA kits (Shanghai ELISA Biotechnology Co., Ltd., Shanghai, China). Of note, the intra- and inter-assay CVs for these ELISA kits were ≤10%.

#### 2.4.2. Hepatic HMGCR and CYP7A1 Activities

Liver samples were frozen immediately in liquid nitrogen. The liver microsomes were prepared by homogenization and centrifugation as described by [16,17]. After that, the supernatant was taken to determine the HMGCR activity by using the Pig HMGCR ELISA Kit (E10451p, EIAAB SCIENCE INC, Wuhan, China) according to the manufacturer’s instruction. The CYP7A1 activities in the liver of pigs were assessed by the Pig CYP7A1 ELISA Kit (E1053p, EIAAB SCIENCE INC, Wuhan, China).

#### 2.4.3. Intestinal Morphology

After 48 h fixation in 10% neutral buffered formalin, the histological samples of ileum were washed, excised, dehydrated, and embedded in the paraffin wax. About five non-successive sections of each histological tissue were sliced 5 μm thick, installed on glass slides, and dyed with hematoxylin and eosin. Villus height and crypt depth were measured by an observer under a light microscope (CK-40, Olympus, Tokyo, Japan) at 40× the objective magnification and analyzed with an Image Analyzer (Lucia Software. Lucia, ZaDrahou, Czechoslovakia). About 10 orientated villi and their adjoining crypts of each slice were randomly selected for the calculation of the average villus height, crypt depth, and the ratio of villus height to crypt depth. The same villus was used to determine the number of intraepithelial lymphocytes and goblet cells, which were counted in the same crypts of ileal sections [18].

#### 2.4.4. Intestinal Microbiota Analysis

The bacterial genomic DNA of colonic digesta was extracted by a Stool DNA Kit (Omega Bio-TEK, Norcross, GA, USA). The sequencing process was conducted as previously described [19]. DNA sequence analysis was carried out by using the pipeline Quantitative Insights Into Microbial Ecology (QIIME) software. Demultiplexing was performed after extracting barcodes from the reads. The Greengenes 13_8 99% operational taxonomic unit (OTU) database was used as the reference database for open-reference OUT picking.

### 2.5. Statistical Analysis

Each pig was considered an experimental unit and all data were analyzed in a randomized complete block design using the Generalized Linear Model procedure of SAS (SAS Institute Inc., Cary, NC, USA). The data were subjected to one-way analysis of variance with post hoc Tukey’s significant difference test. Linear Discriminant Analysis Effect Size (LEfSe) was performed on the BMKCloud platform (www.biocloud.net, accessed on 1 June 2022) with *p* < 0.05 and LDA > 3.0. Values are presented as the least squares means with standard error of the mean (SEM). Significant differences were reported at *p* ≤ 0.05 and trends were noted at 0.05 < *p* ≤ 0.10.

## 3. Results

### 3.1. Plasma Metabolic and Immunological Biomarkers

As shown in Table 1, the plasma HDL contents in the LP, LP + DCA, LP + GLUC and LP + DCA + GLUC groups were higher than in the CON group (*p* < 0.05), and DCA or GLUC supplementation did not significantly change the plasma HDL contents compared with the LP diet (*p* > 0.10). Compared with the CON, the LP diet did not significantly affect the VLDL contents (*p* > 0.10), while DCA along with or without GLUC supplementation induced lower VLDL contents in the plasma of finishing pigs *(p* < 0.05). The LP diet increased the TG concentrations in the plasma of pigs compared with the CON diet (*p* < 0.05), whilst LP + GLUC and LP + DCA + GLUC diets induced lower TG concentrations than the LP diet (*p* < 0.05). The TC concentrations in the plasma of pigs fed with the LP diet were higher than the CON (*p* < 0.05); moreover, pigs fed with the LP + DCA + GLUC diet had lower plasma TC concentrations compared with the LP diet (*p* < 0.05). The LP diet induced a lower TBA concentration in plasma of pigs than the CON diet (*p* < 0.05), and the LP + DCA + GLUC diet elevated the TBA concentration compared with the LP diet (*p* < 0.05). Compared with the CON diet, the LP diet resulted in lower BUN concentrations in the plasma of pigs (*p* < 0.05), and pigs fed with the LP + DCA diet showed higher BUN concentrations than that of the LP diet (*p* < 0.05). The ALB and TP contents in the plasma of pigs were reduced with the LP diet that that of the CON diet (*p* < 0.05), whereas the LP + DCA and LP + GLUC diets increased the ALB content and LP + GLUC diet increased the TP contents compared with the LP diet (*p* < 0.05). The DBil concentrations in the plasma of pigs were higher in the LP group than the CON group (*p* < 0.05) and were lower in the LP + GLUC and LP + DCA + GLUC groups compared with the LP group (*p* < 0.05). The LP diet did not significantly affect the LDH and GGT activities compared with the CON diet (*p* > 0.10), and the DCA or DCA addition did not show significant change in LDH or GGT activities compared with the LP diet (*p* < 0.10). Compared with the CON diet, the LP diet increased the plasma ALP activities, and LP + DCA diet showed a lower ALP activity than the LP diet (*p* < 0.05). The IgG contents in the plasma of finishing pigs were lower in the LP group than the CON group (*p* < 0.05) and were higher in the LP + GLUC group when compared with the LP group (*p* > 0.10). However, there were no significant changes in LDL, GLUC, TBil, PDH, GDH, ADA, ChE, ALT, AST, IgA, and IgM indices between the CON, LP, LP + GLUC, and LP + DCA + GLUC groups (*p* > 0.10).

### 3.2. Hormone Concentrations in Plasma

As shown in Table 2, pigs fed with the LP diet had higher T3 concentrations in the plasma than that of the CON diet, whereas LP + DCA, LP + GLUC, and LP + DCA + GLUC diets significantly increased the T3 concentrations compared with the LP diet (*p* < 0.05). The T4, insulin, and glucagon concentrations in the plasma of the pigs were higher with the LP diet than the CON diet; moreover, the LP + DCA and LP + GLUC, but not LP + DCA + GLUC diets induced higher T4, insulin, and glucagon concentrations compared to the LP diet (*p* < 0.05). The growth hormone concentrations in the plasma of pigs were increased with the LP diet, while LP + DCA, LP + GLUC, and LP + DCA + GLUC diets showed higher growth hormone concentrations compared with the LP diet (*p* < 0.05). Compared with the CON diet, the LP diet showed higher leptin concentrations (*p* < 0.05), and LP + DCA diet increased leptin concentrations compared with the LP diet (*p* < 0.05). The melanin concentration was lower in the plasma of pigs fed with the LP diet than the CON diet, whereas it was higher with the LP + GLUC and LP + DCA + GLUC diets compared with the LP diet (*p* < 0.05).

### 3.3. HMGCR and CYP7A1 in Liver

The enzyme activities involved in cholesterol synthesis in the liver tissue of the finishing pigs are shown in Table 3. The LP diet did not significantly affect the HMGCR content or CYP7A1 activity compared with the CON diet (*p* > 0.10). However, the LP + DCA, LP + GLUC, and LP + DCA + GLUC diets reduced the HMGCR contents in the liver tissues of pigs compared with the LP diet (*p* < 0.05), and the LP + DCA and LP + GLUC diets induced higher CYP7A1 activities than the LP diet (*p* < 0.05).

### 3.4. Ileal Histology

Table 4 shows the ileal histology of finishing pigs. The villus height, crypt depth, ratios of villus height to crypt depth, lymphocytes amount, and goblet cells per crypt were not significantly changed by dietary treatments (*p* > 0.05, Table 4).

### 3.5. Colonic Microbial Diversity and Composition

Figure 1 shows the alpha diversity of colonic microbiota in the finishing pigs. Although the Chao1, Shannon, and Simpson indices were not significantly affected by the dietary treatments (*p* > 0.10), a lower ACE index of the colonic microbiota was shown in the LP + GLUC + DCA group compared with the LP, LP + DCA, and LP + GLUC groups (*p* < 0.05). The analysis of similarities showed that the structure of the colonic microbiota did not significantly differ among dietary treatments (*p* > 0.05, Figure 2a) and PCoA of the weighted Unifrac distance metric revealed no distinct differences in colonic microbiota among groups (Figure 2b). As shown in Figure 3a, the composition of abundant bacteria (Top 10) is provided at genus level in the colon digesta of finishing pigs. The colonic microbiota composition including *Streptococcus*, *Lactobacillus*, *Lachnospiraceae_XPB1014_group* and *Prevotellaceae_NK3B31_group*, *uncultured_bacterium_f_Muribaculaceae*, *Lachnospiraceae_NK4A136_group*, *Rikenellaceae_RC9_gut_group*, *Prevotellaceae_UCG-003*, and *Prevotellaceae_UCG-001* at genus level was not affected by dietary treatments (*p* > 0.10, Table 5). However, the *Ruminococcaceae_UCG_005* was reduced in pigs fed with the LP + DCA + GLUC diet compared with the LP diet (*p* < 0.05). LEfSe analysis was also performed to assess the enrichment of taxa in the four dietary groups (Figure 3b). The consumption of CON diet enriched the *F082*, *Peptococcaceae*, and *Peptococcus*; LP diet led to high relative abundance of *Ruminococcaceae*, *Ruminococcaceae_UCG_005*, *Terrisporobacter*, *Ruminococcus_2*, and *Family_XII_UCG_001*; LP + DCA diet induced high relative abundance of *Lachnospiraceae_UCG_004* and *Ruminococcaceae_UCG_008*; LP + GLUC diet was associated with enriched *Oscillospira, Eubacterium_coprostanoligenes_group*, and *Eubacterium_nodatum_group*; and LP + DCA + GLUC diet showed high relative abundance of *Lachnospiraceae_XPB1014_group*, *Paludibacteraceae*, *Candidatus_Saccharimonas*, *Saccharimonadia*, *Patescibacteria*, and *Saccharimonadales*.

## 4. Discussion

LP diets are commonly used in growing-finishing pigs during the swine production for reducing the nitrogen excretion and saving protein sources. Some researchers have investigated the effects of LP diets on growth performance, carcass traits, and meat quality, while the influence of LP diets on the metabolic biomarkers and intestinal microbiota remains unclear. In this study, DCA and GLUC were added in the LP diet to investigate if they could function as metabolic modulators to increase the glucose or lipid oxidation and reduce the amino acid catabolism, and whether there were changes in intestinal microbiota in finishing pigs.

The present results showed that the LP diet increased the plasma HDL, TG, and TC levels, which was also reported by previous study [20]. DCA supplementation in the LP diet showed limited effects on the lipid profile in the plasma of pigs, whereas DCA along with GLUC supplementation showed greater influence on reducing the VLDL, TG and TC contents. In particular, we found higher TBA concentration in the LP + DCA + GLUC group, which could explain the lower concentrations of VLDL, TG, and TC in the plasma of pigs. We speculated that the higher plasma HDL, TG, and TC levels induced by the LP diet could be associated with the intake of high dietary carbohydrate and fat content in pigs and DCA along with GLUC supplementation might modulate the lipid profile in the plasma of pigs by stimulating the bile acid secretion. Moreover, an LP diet reduced the plasma total protein and urea nitrogen content of pigs. Similar with these findings, some studies also found that an LP diet reduced the plasma total protein and urea nitrogen contents in pigs [21,22,23], which was possibly associated the deficiency of non-essential amino acids and reduced nitrogen excretion; therefore, the amount of deamination and transamination of amino acids decreased in LP diet fed pigs [24]. Meanwhile, the decreased ALB and TP concentrations in the plasma was seen in pigs fed with the LP diet compared with that of the CON diet. This was reasonable since the contents of plasma ALB and TP reflect the dietary protein levels and the degree of digestion, absorption, and utilization of protein by the animals [24]. However, the DCA supplementation in the LP diet might change amino acid metabolism since the increased plasma BUN and ALB content was observed when compared with the LP diet. The activities of liver enzymes, such as ALT, AST, ALP, DBil, or TBil, in plasma were markers of hepatocellular health. The current study showed the DBil content and ALP activity were elevated in the plasma of pigs fed with the LP diet. The DBil was regarded as antioxidant and protective agent against oxidative reactions in liver damage, whilst the ALP is elevated with hepatocellular injuries and an elevation in plasma DBil and ALP indicated the inhibition of bile excretion [25]. Therefore, the increased DBil content and ALP activity in the LP group was possibly associated with liver damage or higher fat accumulation in the liver of pigs, which was also reflected by the decreased plasma TBA concentrations [26]. Furthermore, the lower ALP activity in the LP diet supplemented with DCA might indicate the improvement of liver health by the DCA.

Thyroid hormones, including T3 and T4, play a crucial role in modulating the thermogenic response. Our study showed that the reduction of dietary protein levels increased plasma T3 and T4 concentrations and DCA or GLUC supplementation increased the concentrations further. In the early studies, low-protein high-carbohydrate diets were shown to increase serum T3 and T4 concentrations in pigs or rats [27,28]. The decreased energetic efficiency observed in growing rats and pigs fed LP diets is often associated with a marked rise in T3 and T4 concentrations, which is frequently interpreted as an adaptive diet-induced thermogenesis [28]. In this study, the higher T3 and T4 concentrations could be the results of long-term insufficient intake of non-essential amino acids which altered the thyroid gland hormone secretion. Interestingly, the present study found the LP diets increased growth hormone levels in the plasma of pigs which was contrary to previous findings that diets high in protein increased growth hormone in humans [29]. The increased growth hormone secretion can inhibit protein breakdown and stimulate protein synthesis in muscle and other tissues and inhibit amino acid degradation in liver [29,30]. The elevated growth hormone concentrations especially in the LP groups supplemented with DCA or GLUC were signs of improved protein metabolism in the liver and was also demonstrated by the increased plasma ALB and TP concentrations. Moreover, inconsistent with the previous findings [31], we found the insulin, glucagon, and leptin concentrations in the plasma of finishing pigs were increased with the LP diet compared with the CON diet, which could be the results of increased fat accumulation of pigs induced by a long-term intake of high-carbohydrate and high-fat diet. It was possible that DCA and GLUC promoted the glucose and lipid oxidation, therefore showing greater effects on increasing plasma hormone concentrations [1,32].

In the liver, HMGCR and CYP7A1 are two pivotal enzymes controlling the maintenance of cholesterol homeostasis, wherein the HMGCR is involved in cholesterol de novo biosynthesis and CYP7A1 is involved in cholesterol catabolism [17]. In the present study, DCA or GLUC supplementation in the LP diets reduced the HMGCR content in the liver of pigs. Previous study found that the reduced HMGCR activity led to inhibited de novo synthesis of cholesterol in the liver and, thus, reduced the serum cholesterol level [33]. Furthermore, leptin supplementation has been reported to suppress the mRNA levels of HMGCR in the liver of mice [32]. Similarly, the reduced HMGCR activity in the DCA and GLUC supplemented groups possibly meant the de novo synthesis of cholesterol in the liver was reduced, which was in accordance with the change of plasma leptin concentrations and supported our hypothesis that DCA and GLUC supplementation in an LP diet might regulate the lipid metabolism of pigs. CYP7A1 is the rate-limiting enzyme in the classical bile acid biosynthetic pathway [34], and our study showed that the activity of CYP7A1 in the liver of pigs was increased in the LP diets with DCA or GLUC supplementation. Considering the increased CYP7A1 activity indicated promoted conversion rate of cholesterol into bile acid, DCA or GLUC would be beneficial for increasing the total bile acid synthesis, therefore, to reduce the cholesterol level in the liver [17].

Dietary protein source and composition play important roles in the composition and function of gut microbiota and the LP diet with balanced AA displayed beneficial effects on gut health by modulating the composition and diversity of gut microbiota [21]. The reduction of dietary protein levels did not affect colonic microbial richness or diversity, as shown by Chao1 and ACE species richness, and Shannon and Simpson indices, respectively. Previous studies reported that dietary protein levels had no significant effects on microbiota diversity but altered microbial composition in the hindgut of growing-finishing pigs [5,35,36]. The CON diet showed enriched *F082* and *Peptococcaceae* in the colon of pigs, while previous study reported that the relative abundance of *Peptococcaceae* in the colon of pigs was increased with the high protein diets [19]. The CON diet could induce a relative lower nitrogen utilization efficiency, thus increasing the protein fermentation in the hindgut and the abundance of protein fermentation microbiota such as *Peptococcaceae* [19,37]. However, the role of the Bacteroidetes unclassified family *F082* is still unclear [38]. The LP diet presented increased colonic abundances of *Ruminococcaceae* and *Terrisporobacter*, which was similar with the findings in humans [39]. Both *Ruminococcaceae* and *Terrisporobacter* can degrade and ferment dietary fiber containing complex polysaccharides; therefore, the enriched *Ruminococcaceae* and *Terrisporobacter* abundance could be the results of high carbohydrate content in the LP diet. With the DCA supplementation, the LP diet enriched *Lachnospiraceae* and *Ruminococcaceae* in the colon of pigs. Meanwhile, the GLUC supplementation in the LP diet enriched the *Oscillospira*, *Ruminococcaceae*, and *Eubacterium* abundance in the colon of pigs. This was also seen in a previous study [35,36], and a possible reason could be the high content of corn in the LP diet since *Lachnospiraceae* and *Ruminococcaceae* and *Oscillospira* are polysaccharide-degrading bacterium [40]. These enriched bacteria can produce SCFA by fermenting carbohydrates, in particular, directly or indirectly exhibits positive regulatory effects on obesity and chronic inflammation [41]. With DCA along with GLUC addition, the relative abundance of *Lachnospiraceae_XPB1014_group* and *Lachnospiraceae_UGG_004* was enriched. This was reported by a previous study which studied the effects of an LP diet supplemented with alpha-ketoglutarate on cecal microbial community in growing pigs [42]. Similar to alpha-ketoglutarate, DCA might also serve as a metabolic fuel for the gut in pigs fed an LP diet and improve the microbial fermentation. Furthermore, the LP diet supplemented with DCA along with GLUC increased the relative abundance of *Paludibacteraceae*. This was also seen in a previous study that the LP + BCAA diet showed higher abundance of *Paludibacteraceae* in the feces of nursery pigs compared to the CON diet [43]. The enriched *Paludibacteraceae* could be attributed to the high dietary carbohydrate content which was used as a fermenting substrate by the bacteria [44]. However, it should be noticed that the safety of DCA is very important for both pigs and human. DCA is a medicine used to treat cancer in humans, but the effects of DCA on pigs were lesser known; therefore, the safety in pig production still need further investigation. Moreover, it remains important to identify more economic additives that function as DCA for pig industry.

## 5. Conclusions

In conclusion, an LP diet increases the concentrations of lipids and hormones related to energy metabolism in the plasma of finishing pigs, but did not affect the intestinal morphology and microbial diversity of pigs. The 120 mg/kg DCA or 1.8% GLUC supplementation to an LP diet shows limited effects on the lipid profile in the plasma of pigs, but modulates the hepatic cholesterol metabolism by showing a lower HMGCR content and higher CYP7A1 activity. Moreover, the DCA along with GLUC supplementation mainly improves the plasma lipid profile and stimulates the bile acid secretion. The reduction of dietary protein level, and DCA and GLUC supplementation changed the colonic microbiota composition, mainly due to the high dietary carbohydrate content. Overall, these findings suggest that DCA or GLUC supplementation in an LP diet can modulate the lipid metabolism and increase the relative abundance of bacteria involved in carbohydrate fermentation in finishing pigs.

## Figures and Tables

**Figure 1 animals-12-02522-f001:**
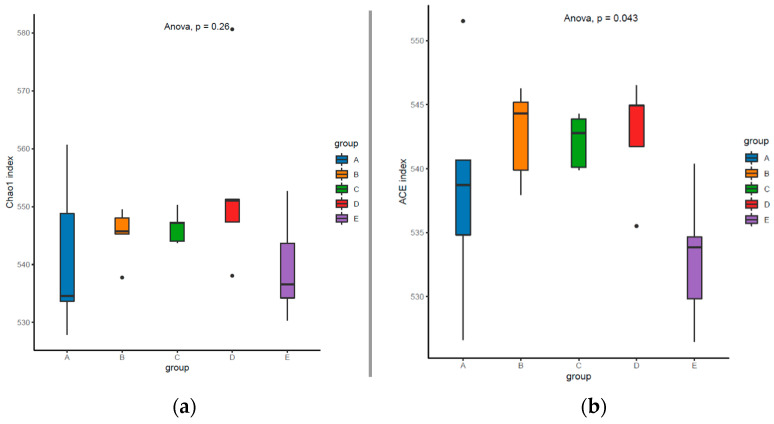
Alpha diversity including Chao1 (**a**), Ace (**b**), Shannon (**c**), and Simpson (**d**) of the colon communities of individual pigs after feeding control diet (A, CON), low protein diet (B, LP), LP + sodium dichloroacetate diet (C, LP + DCA), LP + sodium dichloroacetate diet (D, LP + GLUC), and LP + DCA + GLUC diet (E), respectively. *n* = 5.

**Figure 2 animals-12-02522-f002:**
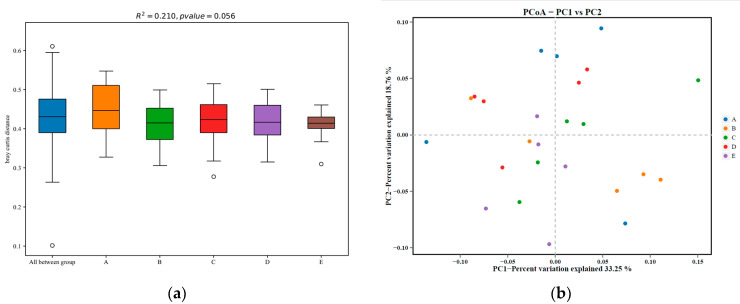
The analysis of similarities (ANOSIM) tests of gut microbiota performed between groups based on relative abundance of OTU (**a**) and principal coordinate analysis plot of the weighted Unifrac metric (**b**) of the colon communities of individual pigs after feeding control diet (A, CON), low protein diet (B, LP), LP + sodium dichloroacetate diet (C, LP + DCA), LP + sodium dichloroacetate diet (D, LP + GLUC), and LP + DCA + GLUC diet (E), respectively. *n* = 5.

**Figure 3 animals-12-02522-f003:**
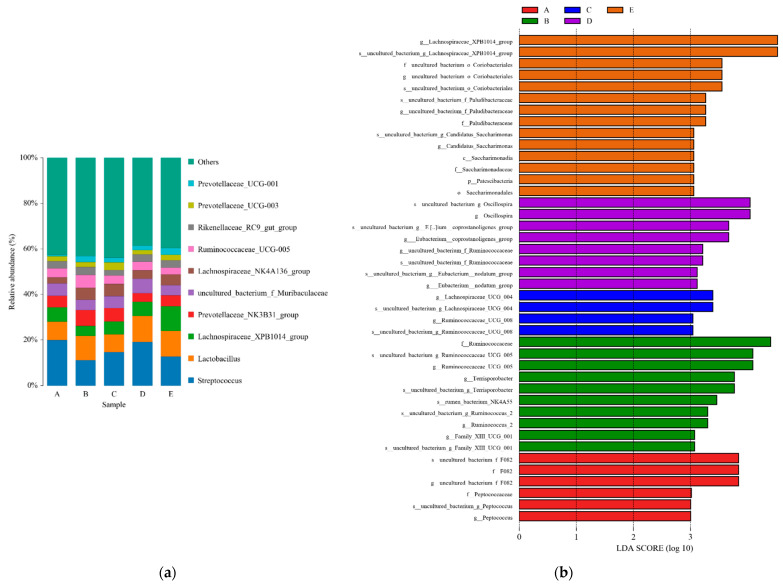
Microbiota composition at genus level (**a**) and linear discriminant analysis effect size analysis (LEfSe) (**b**) of the colon communities of individual pigs after feeding control diet (A, CON), low protein diet (B, LP), LP + sodium dichloroacetate diet (C, LP + DCA), LP + sodium dichloroacetate diet (D, LP + GLUC), and LP + DCA + GLUC diet (E), respectively. *n* = 5. LEfSe analysis (LDA > 3), showing the most differentially abundant taxa enriched in microbiota from pigs.

**Table 1 animals-12-02522-t001:** Effects of supplementing low protein (LP) diets with sodium dichloroacetate (DCA) and glucose (GLUC) on the plasma metabolic and immunological biomarkers of growing-finishing pigs.

Item	Treatments					SEM	*p*-Value
	CON	LP	LP + DCA	LP + GLUC	LP + DCA + GLUC		
HDL (mg/mL)	0.98 ^b^	1.34 ^a^	1.31 ^a^	1.25 ^a^	1.20 ^a^	0.05	<0.001
LDL (mg/mL)	0.47	0.47	0.56	0.52	0.52	0.03	0.104
VLDL (mg/mL)	6.97 ^ab^	10.42 ^a^	4.95 ^b^	6.95 ^ab^	4.33 ^b^	1.30	0.026
TG (mmol/L)	1.48 ^b^	1.88 ^a^	1.97 ^a^	1.51 ^b^	1.15 ^c^	0.07	<0.001
TC (mmol/L)	4.20 ^d^	6.28 ^ab^	6.49 ^a^	5.42 ^bc^	5.18 ^c^	0.30	<0.001
GLUC (mmol/L)	17.9	19.0	20.0	23.9	18.1	2.36	0.388
TBA (μmol/L)	37.3 ^b^	27.7 ^c^	32.6 ^bc^	27.8 ^c^	42.5 ^a^	1.73	<0.001
BUN (mmol/L)	5.33 ^a^	3.97 ^b^	4.88 ^a^	3.51 ^b^	3.53 ^b^	0.29	<0.001
ALB (g/L)	39.2 ^a^	37.1 ^b^	40.0 ^a^	39.3 ^a^	38.9 ^ab^	0.49	0.017
TP (mg/mL)	95.4 ^a^	78.1 ^b^	90.2 ^ab^	95.4 ^a^	77.5 ^b^	3.95	0.004
DBil (μmol/L)	1.93 ^b^	2.72 ^a^	2.69 ^a^	0.89 ^c^	1.31 ^bc^	0.21	<0.001
TBil (μmol/L)	13.1	13.3	10.6	14.7	11.3	1.33	0.226
LDH (U/L)	491 ^a^	467 ^ab^	393 ^b^	460 ^ab^	450 ^ab^	20.7	0.032
PDH (U/L)	16.4	34.3	19.7	25.1	16.0	3.33	0.194
GDH (U/L)	2.49	2.93	2.27	2.35	2.26	0.32	0.564
ADA (U/L)	1.79	1.56	2.31	1.85	1.78	0.25	0.323
ChE (U/L)	305	333	238	276	316	36.3	0.398
GGT (U/L)	28.1 ^b^	30.5 ^ab^	36.2 ^a^	27.0 ^b^	32.5 ^ab^	1.76	0.008
ALP (U/L)	12.8 ^c^	19.8 ^a^	16.6 ^b^	17.2 ^ab^	18.6 ^ab^	0.77	<0.001
ALT (U/L)	13.7	12.3	14.7	15.0	12.7	1.38	0.550
AST (U/L)	10.7	10.7	9.45	12.7	11.6	1.19	0.420
IgA (μg/mL)	403	385	381	383	433	63.0	0.973
IgG (μg/mL)	9.35 ^a^	4.85 ^b^	7.79 ^ab^	9.86 ^a^	7.38 ^ab^	0.85	0.003
IgM (μg/mL)	5.94	6.05	6.29	5.21	6.38	1.36	0.976

Data are presented as means (*n* = 6). ^a,b,c,d^ Values within a row with different superscripts differ significantly (*p* < 0.05). Abbreviations: ADA, adenosine deaminase; ALB, albumin; ALP, alkaline phosphatase; ALT, alkaline phosphatase; AST, aspartate transaminase; BUN, blood urea nitrogen; ChE, cholinesterase; CON, control group; DBil, direct bilirubin; DCA, sodium dichloroacetate; GDH, glutamate dehydrogenase; GGT, γ-glutamyltransferase; GLUC, glucose; HDL, high-density lipoprotein; IgA, immunoglobulin A; IgG, immunoglobulin G; IgM, immunoglobulin M; LDH, lactate dehydrogenase; LDL, low-density lipoprotein; PDH, pyruvate dehydrogenase; TBA, total bile acids; TBil, total bilirubin; TC, total cholesterol; TG, triacylglycerol; TP, total protein; VLDL, very low-density lipoprotein.

**Table 2 animals-12-02522-t002:** Effects of supplementing low protein (LP) diets with sodium dichloroacetate (DCA) and glucose (GLUC) on the plasma hormone concentrations in growing-finishing pigs.

Item	Treatments					SEM	*p*-Value
	CON	LP	LP + DCA	LP + GLUC	LP + DCA + GLUC		
T3 (pmol/L)	1.93 ^c^	2.81 ^b^	3.93 ^a^	4.14 ^a^	3.95 ^a^	0.19	<0.001
T4 (pmol/L)	3.01 ^c^	5.68 ^b^	7.96 ^a^	7.49 ^a^	5.74 ^b^	0.47	<0.001
Growth hormone (ng/mL)	2.26 ^d^	4.68 ^c^	7.18 ^a^	7.65 ^a^	5.95 ^b^	0.36	<0.001
Insulin (mIU/L)	2.71 ^c^	4.67 ^b^	8.38 ^a^	7.81 ^a^	4.79 ^b^	0.21	<0.001
Glucagon (pg/mL)	30.6 ^c^	51.4 ^b^	76.4 ^a^	69.3 ^a^	45.5 ^b^	2.88	<0.001
Leptin (ng/mL)	2.03 ^c^	2.61 ^b^	3.18 ^a^	2.94 ^ab^	2.72 ^b^	0.10	<0.001
Melanin (pg/mL)	119 ^a^	66.7 ^b^	68.0 ^b^	114 ^a^	100 ^a^	6.02	<0.001

Data are presented as means (*n* = 6). ^a,b,c,d^ Values within a row with different superscripts differ significantly (*p* < 0.05). Abbreviations: CON, control group.

**Table 3 animals-12-02522-t003:** Effects of supplementing low protein (LP) diets with sodium dichloroacetate (DCA) and glucose (GLUC) on the enzyme activities in liver tissue of growing-finishing pigs.

Items	Treatments					SEM	*p*-Value
	CON	LP	LP + DCA	LP + GLUC	LP + DCA + GLUC		
HMGCR (ng/mL)	238 ^a^	230 ^a^	161 ^b^	173 ^b^	198 ^b^	10.4	<0.001
CYP7A1 (U/L)	0.02 ^d^	0.03 ^cd^	0.05 ^a^	0.04 ^b^	0.03 ^bc^	0.001	<0.001

Data are presented as means (*n* = 6). ^a,b,c,d^ Values within a row with different superscripts differ significantly (*p* < 0.05). Abbreviations: CON, control group.

**Table 4 animals-12-02522-t004:** Effects of supplementing low protein (LP) diets with sodium dichloroacetate (DCA) and glucose (GLUC) on the ileum histology of growing-finishing pigs.

Item	Treatments					SEM	*p*-Value
	CON	LP	LP + DCA	LP + GLUC	LP + DCA + GLUC		
Villus height (μm)	495	509	409	447	353	47.2	0.164
Crypt depth (μm)	302	318	289	289	257	27.6	0.623
Villus height/Crypt depth	1.64	1.63	1.54	1.60	1.41	0.19	0.911
Lymphocytes amount per villus	72.6	61.8	56.8	79.2	97.8	19.2	0.600
Goblet cells per crypt	10.2	10.4	13.4	5.2	3.2	2.47	0.051

Data are presented as means (*n* = 6). Abbreviations: CON, control group.

**Table 5 animals-12-02522-t005:** Effects of supplementing low protein (LP) diets with sodium dichloroacetate (DCA) and glucose (GLUC) on colonic microbiota composition at genus level (Top 10) in growing-finishing pigs.

Item	Treatments					SEM	*p*-Value
	CON	LP	LP + DCA	LP + GLUC	LP + DCA + GLUC		
*Streptococcus*	0.19	0.11	0.14	0.19	0.13	0.08	0.370
*Lactobacillus*	0.08	0.11	0.08	0.12	0.11	0.05	0.697
*Lachnospiraceae_XPB1014_group*	0.06	0.04	0.06	0.16	0.11	0.04	0.127
*Prevotellaceae_NK3B31_group*	0.02	0.02	0.03	0.02	0.02	0.007	0.072
*uncultured_bacterium_f_Muribaculaceae*	0.05	0.04	0.05	0.06	0.04	0.03	0.721
*Lachnospiraceae_NK4A136_group*	0.03	0.05	0.05	0.04	0.05	0.02	0.228
*Ruminococcaceae_UCG-005*	0.04 ^ab^	0.05 ^a^	0.04 ^ab^	0.04 ^ab^	0.03 ^b^	0.01	0.048
*Rikenellaceae_RC9_gut_group*	0.03	0.03	0.02	0.03	0.03	0.01	0.693
*Prevotellaceae_UCG-003*	0.02	0.02	0.03	0.02	0.02	0.007	0.072
*Prevotellaceae_UCG-001*	0.05	0.03	0.02	0.02	0.03	0.02	0.264

Data are presented as means (*n* = 5). ^a,b^ Values within a row with different superscripts differ significantly (*p* < 0.05). Abbreviations: CON, control group.

## Data Availability

The data presented in this study are available on request from the corresponding author.

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
