# Peer review of "Effects of Long-Term Low-Protein Diets Supplemented with Sodium Dichloroacetate and Glucose on Metabolic Biomarkers and Intestinal Microbiota of Finishing Pigs"

_animals, 2022, doi:10.3390/ani12192522_

Round 1

Reviewer 1 Report

This manuscript targets to evaluate the effects of long-term low-protein diets supplemented with sodium dichloroacetate and glucose on metabolic response and intestinal microbiota of finishing pigs. The exploration about sodium dichloroacetate in low protein diet is novel, but there are still some questions need to be answered. 

1.     In the M&M, the analysis method of hepatic enzymes was not very clear.

2.     How can you distinguish the goblet cells with HE method?

3.     How was the dosage of DCA determined? Did these pigs enter the food supply since the DCA might be toxic to pigs?

4.     What’s the additional feed cost associated with the addition of DCA?

Author Response

Response to Reviewer 1

Reviewer(s)' Comments to Author: This manuscript targets to evaluate the effects of long-term low-protein diets supplemented with sodium dichloroacetate and glucose on metabolic response and intestinal microbiota of finishing pigs. The exploration about sodium dichloroacetate in low protein diet is novel, but there are still some questions need to be answered.

  1. In the M&M, the analysis method of hepatic enzymes was not very clear.

Response: Thanks for your suggestion, we have corrected the analysis method of the two hepatic enzyme activity.

  1. How can you distinguish the goblet cells with HE method?

Response: The goblet cells can be distinguished and counted the H&E stained sections under a Motic Med 6.0 CMIAS Image Analysis System (Motic China Group Co., Ltd., Guangzhou, China). We here listed two reference which also used this method (Wu et al., 2021. Effects of Glutamine on Lymphocyte Proliferation and Intestinal Mucosal Immune Response in Heat-Stressed Broilers; Rodrigo et al., 2014. Histological and histochemical features of the small intestine in loggerhead sea turtle hatchlings (Caretta caretta).

  1. How was the dosage of DCA determined? Did these pigs enter the food supply since the DCA might be toxic to pigs?

Response: The dosage of DCA was chosen based on our previous study and was adjusted from a pretest which also used growing pigs and showed better growth performance with 120 mg/kg DCA addition. This information has been added in our previous study which reported the effects of DCA on the growth performance and carcass quality of growing-finishing pigs (Xu et al., 2022. Effects of supplementing low-protein diets with sodium dichloroacetate and glucose on growth performance, carcass traits, and meat quality of growing-finishing pigs). And we totally agree that the safety of DCA is very important for both pigs and human. DCA is a medicine used to treat cancer in human but the effects of DCA on pigs were less known therefore the safety in pig production still need further investigation. Pigs for sampling and fed with DCA did not enter the food supply and was handled based on the experimental animal handling procedure of Southwest University. We added this message in the manuscript in line 409-412.

  1. What’s the additional feed cost associated with the addition of DCA?

Response: This is a good point. DCA is about 313 dollars/kg and the addition of DCA in LP diets would increase the feed cost compared with other feed additives. However, we used DCA as a metabolic modulator to investigate the metabolic mechanism, and we hope we can identify a kind of more economic addictive that works like DCA for pig industry or reduce the cost of DCA by improving its production process. We have clarified this in Discussion part, please see line 412-413.

Reviewer 2 Report

This study by Xu et al explores the effects of long-term low-protein diets supplemented with sodium dichloroacetate and glucose on metabolic response and intestinal microbiota of finishing pigs. Overall, the experiment was well conducted and the English expression was also clear. I only have few minor comments.

Comments to the Author

1.     Why only male pigs were chosen for the experiment?

2.     Did you do the power calculation for the experiment?

3.     In the discussion part, you mentioned the changes of plasma metabolic biomarkers were possibly induced by the high carbohydrate content in the LP diets. However, as I can see, the fat content in the LP diets was also increased than the control diet.

4.     Is the DCA bitter? Does it change the feed intake of pigs due to its flavor?

5.     Line 105 10-mL should be 10 mL.

6.     Make sure if there is space between number and temperature degree unit.

Author Response

Response to Reviewer 2

Reviewer(s)' Comments to Author: This study by Xu et al explores the effects of long-term low-protein diets supplemented with sodium dichloroacetate and glucose on metabolic response and intestinal microbiota of finishing pigs. Overall, the experiment was well conducted and the English expression was also clear. I only have few minor comments.

  1. Why only male pigs were chosen for the experiment?

Response: Female and male pigs had different effects on the protein deposition with the low protein diets. Therefore, we only chose male pigs to reduce the deviation. However, it’s still important to use female pigs to carry out the experiment, which need to be investigated in the future.

  1. Did you do the power calculation for the experiment?

Response: The power calculation was performed in this study, however, we did not declare it in this manuscript since we have already describe it in our last paper (Xu et al., 2022. Effects of supplementing low-protein diets with sodium dichloroacetate and glucose on growth performance, carcass traits, and meat quality of growing-finishing pigs).

  1. In the discussion part, you mentioned the changes of plasma metabolic biomarkers were possibly induced by the high carbohydrate content in the LP diets. However, as I can see, the fat content in the LP diets was also increased than the control diet.

Response: This is a good suggestion. We revised it in the discussion part. Please see line 311 and line 354.

  1. Is the DCA bitter? Does it change the feed intake of pigs due to its flavor?

Response: The DCA does have an irritating smell. However, the dosage of DCA addition is small and did not significantly change the feed intake of growing-finishing pigs at three growing stages.

  1. Line 105 10-mL should be 10 mL.

Response: Corrected.

  1. Make sure if there is space between number and temperature degree unit.

Response: There is space between number and temperature degree unit. We have unified the format of temperature.
